# Associations between Cognitive Impairment, Weight Status and Comorbid Conditions in Hospitalized Adults of 55 Years and Older in Guadeloupe

**DOI:** 10.3390/healthcare12171712

**Published:** 2024-08-27

**Authors:** Livy Nicolas, Valerie Bassien-Capsa, Yann Ancedy, Vaneva Chingan-Martino, Jean-Pierre Clotilde, Yaovi Mignazonzon Afassinou, Olivier Galantine, Rosan Fanhan, Maturin Tabué-Teguo, Lydia Foucan

**Affiliations:** 1Medical Unit, Médical Centre Lucien NICOLAS, Clinique Nouvelles Eaux Marines, Le Moule 97160, Guadeloupe; l.nicolas@clinique-eauxmarines.com (L.N.); yann.ancedy@chu-guadeloupe.fr (Y.A.); jp.clotilde@clinique-eauxmarines.com (J.-P.C.); rosan.fanhan@wanadoo.fr (R.F.); 2Research Team on Cardiometabolic Risk ECM, University Hospital, University of the Antilles, Pointe-à-Pitre 97157, Guadeloupe; valerie.bassiencapsa@chu-guadeloupe.fr (V.B.-C.); togbericardo@yahoo.fr (Y.M.A.); olivier.galantine@chu-guadeloupe.fr (O.G.); 3Cardiology Unit, University Hospital, University of the Antilles, Pointe-à-Pitre 97157, Guadeloupe; 4Diabetic Foot Unit, University Hospital, University of the Antilles, Pointe-à-Pitre 97157, Guadeloupe; vaneva.martino@chu-guadeloupe.fr; 5Laboratoire de Mathématique Informatique et Applications LAMIA (EA 4540), University of the Antilles, Pointe-à-Pitre 97157, Guadeloupe; tabue.maturin@gmail.com; 6Clinical Research Unit, Médical Centre Lucien NICOLAS, Clinique Nouvelles Eaux Marines, Le Moule 97160, Guadeloupe

**Keywords:** aging, chronic conditions, cognitive impairment, obesity, underweight, undernutrition

## Abstract

Cognitive decline and comorbid conditions commonly co-occur, and these conditions can affect cognitive health. We aimed to estimate the prevalence of cognitive impairment (CI) according to weight status and to evaluate the associations between CI, weight status and comorbid conditions in adults of 55 years and older. The Abbreviated Mental Test Score (AMTS) was used. Logistic regressions were performed. Overall, 415 individuals were included. The mean age was 75.7 ± 10.1 years, and the mean BMI was 26.2 ± 6.9 kg/m^2^. The prevalence of CI was 20.7% in the whole study group and 31%, 24.8%, 17.7% and 10.2% in underweight, normal weight, overweight and obese individuals, respectively; *p* < 0.004. The low folate, vitamin D and prealbumin levels were more frequently found in individuals with CI compared with those without CI. Compared with the obese individuals, a higher odds ratio of prevalent CI was noted for underweight individuals OR 3.89 (95% CI 1.54–9.80); *p* = 0.004. Additionally, male gender, older age, stroke, having three or more comorbid conditions and findings of undernutrition were significantly associated with CI. Being underweight was associated with an increased risk of CI. Prevention strategies including the monitoring of nutritional status may help to prevent cognitive decline and promote healthy aging.

## 1. Introduction

The prevalence of both cognitive decline and chronic conditions increase with age. In addition to age-related changes in the brain, multiple other factors can affect cognitive health. Several studies found associations between cognitive impairment (CI) and chronic conditions such as heart disease and stroke [1,2,3], diabetes [4] or chronic kidney disease [5].

Studies have also demonstrated, in various ethnic groups, association between body mass index (BMI) and cognitive function in older adults [6,7,8,9]. A positive association between obesity in midlife and later dementia has been reported, whereas the opposite was found in late life, in a meta-analysis [9].

Various tools can identify CI, which can also be self-perceived, not measurable by clinical testing but, nonetheless, at an elevated risk of dementia [10].

In the brain, in mild CI and Alzheimer’s disease (AD), structural magnetic resonance imaging (MRI) showed brain atrophy and other static tissue abnormalities, which advanced with the progression of the neuro-cognitive disease [11]. Comorbid chronic conditions can induce additional lesions in the brain [12,13].

Cognitive impairment, which can lead to dementia, negatively impacts the health of the elderly, notably with a deleterious effect on the independence of the individuals. It also negatively affects families, communities and health-care systems [14] and can be a burden for caregivers. Dementia prevention, intervention and care will improve living and dying for individuals with dementia and their families [15].

Since there are currently no effective treatments for dementia, it appears necessary to identify potential factors for CI, in particular, modifiable factors, in order to prevent or delay the progression of dementia.

Cardiovascular risk factors, including hypertension, diabetes and abnormal weight status, are modifiable risk factors for CI. They are promising for interventions on dementia. In the island of Guadeloupe (FWI), which has about 375,845 inhabitants, with the majority of individuals being of African descent and exhibiting a very high prevalence of hypertension (32%) [16], diabetes (12%) [17] or obesity (23%) [16], no study, to our knowledge, has reported the prevalence of CI and/or the comorbidities associated with this pathology.

Therefore, in the present study, conducted in hospitalized adults of 55 years and older, we aimed to estimate the prevalence of CI according to weight status and to evaluate the associations between CI, weight status and comorbid conditions.

## 2. Materials and Methods

In a cross-sectional study, between 2021 and 2023, we considered voluntary patients, of both sexes, aged 55 years and older, who were admitted to the medical unit in a health establishment in Guadeloupe. All the participants had benefited from an assessment of their cognitive function with the Abbreviated Mental Test Score (AMTS) [18].

We excluded patients with known Alzheimer’s disease (AD) or dementia, those with missing BMI and those with missing values for the cognitive test. The final sample size was 415 individuals.

The protocol of this study was approved by the ethics committee of the Lucien NICOLAS medical center (Clinique Nouvelles Eaux Marines), No. CE-20210205-01-2021, and by the ethics committee of the Sud-Mediteranee IV, France, No. 20.12.01. This study was registered at the French National Agency for Medicine and Health Products Safety Security (internal reference ANSM-RCB-PAR 1602697118, National No. 2020-A02851-38). The study was conducted in accordance with principles specified in the Declaration of Helsinki. Written informed consent to participate in this study was obtained from all participants.

### 2.1. Data Collection

We collected data including age, personal cardiovascular medical history, treatment at entry, use of antihypertensive or antidiabetic treatments and biological data at entry.

Height and weight were measured with participants standing without shoes and being lightly clothed.

Body mass index (BMI) was calculated as weight/height^2^ (kg/m^2^).

The measurements were made by trained nurses and physicians. Blood pressure was measured according to a standardized protocol with automatic sphygmomanometers.

Laboratory measures: Blood samples were obtained from participants after overnight fasting. Laboratory values were measured by automated and standardized methods and referred to single measures. Serum albumin, serum prealbumin (PAB), folates, vitamin D, vitamin B12 and serum creatinine concentrations were determined

The AMTS was used to identify the presence of CI. This test is a 10-point assessment, [18] introduced to rapidly assess elderly patients for the possibility of dementia. This test is also a quick and easy way for determining the presence of CI in elderly patients, but it is not a substitute for a full cognitive assessment.

This test [18] consists of 10 questions that assess the patient’s orientation, memory and attention. The questions include asking the patient their age, the current year and the name of the current president. The test also includes a question that requires the patient to count backwards from 20 to 1. We considered a score <6 to identify CI with this AMTS test.

### 2.2. Clinical Parameters

Cognitive impairment: the individuals with CI were those having an AMTS score < 6 and those who experience persistent memory loss or symptoms of confusion, noted in the medical report.

Weight status: the individuals were categorized according to their BMI in the following four groups: BMI < 21 kg/m^2^ (underweight), BMI ≥ 21 and <25 kg/m^2^ (normal weight), BMI ≥ 25 and <30 kg/m^2^ (overweight) and BMI ≥ 30 (obesity).

Diabetes–high fasting blood glucose (FBG): known diabetes or FBG ≥ 5.6 mmol/L (100 mg/dL).

Hypertension: systolic blood pressure (SBP) ≥ 140 or a diastolic blood pressure (DBP) ≥ 90 mmHg or a history of hypertension and a current use of antihypertensive medication.

Heart failure: a documented history of cardiac insufficiency.

Coronary artery disease (CAD): diagnosed by physician, angina pectoris, myocardial infarction or coronary artery bypass. Major cardiovascular disease combining CAD, heart failure and history of stroke.

Chronic renal failure (CRF): serum creatinine levels > 1 mg/dL (88.40 µmol/L or 10 µg/mL).

Biological parameters of undernutrition including prealbumin (PAB) level < 0.20 g/L, vitamin D level < 20 ng/mL, folate level < 4.89 ng/mL and vitamin B-12 < 200 pg/mL.

Comorbid conditions, taken into account for the purpose of the present study, were the following: (i) hypertension; (ii) diabetes; (iii) history of stroke; (iv) chronic renal failure; and (v) PAB level < 0.20 g/L (as a biomarker of undernutrition).

### 2.3. Statistical Analysis

Characteristics of the study participants are reported as mean (SDs) for continuous variables and numbers (percentages) for categorical variables.

The chi-squared test was used to test percentage differences between the individuals with and without CI. To test mean differences, we used the analysis of variance (ANOVA).

We explored the potential association between CI and weight status in the individuals with complete data for the co-variables. Two models for logistic regression analysis were performed. Model 1 with two categories for weight status (underweight: Yes/No) and Model 2 with four categories (underweight, normal weight, overweight and obesity). The models were adjusted for age, gender, hypertension, diabetes–high FBG, history of stroke, chronic renal failure and PAB level < 0.20 g/L.

Adjusted odds ratios (ORs) and 95% confidence intervals (95% CIs) were estimated.

The IBM SPSS Statistics software version 21 was used for data analyses. All tests were two-sided, and a *p*-value < 0.05 was considered significant.

## 3. Results

Overall, data of 415 hospitalized individuals with available results for the AMTS test were considered for study.

The mean age of the study population was 75.7 ± 10.1 years (range 55 to 98 years); 51.8% were women. The mean BMI was 26.2 ± 6.9 kg/m^2^.

Among the whole study group, we noted the following comorbidities: 81.7% for hypertension and 32.8%, 21.2%, 41.3% and 8.6% for diabetes–high FBG, obesity, renal failure and history of stroke, respectively.

The prevalence of CI was 20.7% in the whole study group and 25.5% in men vs. 16.3% in women; *p* = 0.021. According to the weight status, we noted 31%, 24.8%, 17.7% and 10.2% among underweight (n = 84), normal weight (n = 113), overweight (n = 130) and obese (n = 88) individuals, respectively; *p* < 0.004; Figure 1.

Among the 86 individuals with CI, 54.7% had an AMTS score lower than 6. The other individuals reported worsening memory or findings of confusion that were noted in their medical hospitalization report.

Characteristics of patients according to the presence/absence of cognitive impairment are presented in Table 1.

No significant differences were found between individuals with and without cognitive impairment for the prevalence of hypertension, diabetes, chronic renal failure and major cardiovascular disease.

Those with cognitive impairment were more likely to be men (*p* = 0.021), aged 75 and over (*p* < 0.001), underweight (*p* = 0.018), with a history of stroke (*p* = 0.019), a prealbumin level < 0.20 g/L (*p* = 0.003) and other findings of undernutrition, had at least three comorbid chronic conditions (*p* = 0.015) and were less likely to be obese (*p* = 0.006).

Among the individuals with diabetes–high FBG, no significant difference was found in those with CI and those without CI for fasting blood glucose and A1C Hb levels.

Considering the distribution of these comorbid conditions according to age categories (age < 75 years/≥ 75 years) (Table 2), differences between individuals with and without CI were noted only for the prevalence of a history of stroke, PAB level < 0.20 g/L, the number of comorbid conditions and only in the individuals younger than 75 years.

### Multivariate Logistic Regression for Prevalent Cognitive Impairment

Table 3 presents the odds ratios for experiencing CI in the 415 individuals with complete data for the co-variables.

Considering two categories (non-underweight/underweight) for weight status, the factors associated with CI were age ≥ 75 years (*p* = 0.001), gender (*p* = 0.042), underweight (*p* = 0.046), history of stroke (*p* = 0.028) and PAB level < 0.20 g/L (*p* = 0.028). Underweight individuals had a higher odds ratio for having CI than the other individuals; OR 1.85 (95% IC 1.01–3.39); *p* = 0.046 (Model 1). Considering four categories (underweight, normal weight, overweight and obesity) for weight status and with obesity as the reference group, the factors associated with CI were age ≥ 75 years (*p* = 0.003), normal weight (*p* = 0.044), underweight (*p* = 0.004), history of stroke (*p* = 0.030) and PAB levels < 0.20 g/L (*p* = 0.044). Thus, compared with the obese individuals, normal weight and underweight individuals had a higher odds ratio of prevalent CI with OR 2.48 (95% CI 1.03–6.01), *p* = 0.044, for normal weight individuals; and 3.89 (95% CI 1.54–9.80), *p* = 0.004, for underweight individuals.

## 4. Discussion

In the present study, in hospitalized Caribbean individuals aged 55 and more years, we investigated the relations between prevalent CI, weight status and comorbid conditions. The prevalence of CI, excluding AD and dementia, was 20.7%. We found that underweight individuals had a significantly increased odds ratio of CI compared to non-underweight individuals or to obese individuals. Additionally, male gender, older age, history of stroke, findings of undernutrition and having three or more comorbid conditions were found to be significantly associated with CI.

The effects of obesity on CI have been previously reported, and the results depended on life stage [7]. In midlife, obesity has been associated with an increased rate of the progression of vascular brain injury, global and hippocampal atrophy, and decline in executive function a decade later [7]. A systematic review and meta-analysis [9] suggest a positive association between obesity in midlife and later dementia but the opposite in late life [9].

The results in our study also highlight the deleterious impact of undernutrition and emphasize the importance, already reported [19], of a good nutritional status for cognitive decline prevention in the elderly [19].

### 4.1. Prevalence of Cognitive Impairment and Gender Differences

The overall prevalence of CI in our participants, 20.7%, was consistent with what has been reported in a systematic review of adults older than 50 years of age [20], with prevalence (80 studies) ranging between 5.1% and 41% and with a median of 19.0% [20].

The prevalence of CI was observed higher in men than in women in our study. Similar results were found in other studies [21,22]. However, women were also reported for having a higher prevalence of CI than men [23].

These differences might be related to different health status, comorbid conditions, ethnicity or education. Moreover, results of a meta-analysis in which fifty-six studies were included found no statistically significant sex differences in the prevalence or incidence of amnestic mild CI [24]. Additionally, there was a significantly higher prevalence, but not incidence, of non-amnestic mild CI among women and no sex differences in studies that combined both subtypes of mild CI [24]. The authors suggested that studies must better characterize the etiology of CI to better understand sex differences in the preclinical stages of dementia [24].

### 4.2. Diabetes, Hypertension and Cognitive Impairment

We found no significant difference between the individuals with CI and those without CI for diabetes–high FBG. Nevertheless, only in the individuals of age < 75 years, we noted a non-significant trend of higher prevalence of this metabolic disease. Considering the individuals with diabetes–high FBG, we did not find any differences concerning A1cHb levels in those with and without CI.

Type 2 diabetes may cause injury to the brain, which could manifest as CI [13]. On the physio-pathological level, there are links between type 2 diabetes and dementia, which are both characterized by metabolic perturbations in the brain including insulin resistance and altered glucose utilization [13]. Diabetes has been reported to be independently associated with CI [13,25].

The absence of a strong association between CI and diabetes in our study might be explained by the fact that individuals with AD or dementia were not included in the present study. According to some authors, cognitive decline in diabetes is progressive [4] and initially subtle, and, in progressive patients, it develops into mild CI followed by frank dementia [4].

In this study population, which had a very high prevalence of hypertension (81.7%), we did not find any significant association between CI and hypertension. However, vascular contributions to CI and dementia in later life are common, and hypertension has been reported as a major risk factor for CI by some authors [26]. However, the results of studies on this association are also controversial when considering age categories and gender. Low diastolic blood pressure was previously associated with a higher risk of dementia in elderly individuals over age 75, and dementia risk was found to be higher in subjects with persistently low blood pressure [27]. In the oldest old (age 85 year), higher systolic blood pressure (SBP) was associated with resilience to cognitive decline [28]. In the Framingham heart study, with a prospective design, adverse effects of obesity and hypertension on cognitive performance were observed for men only [29].

More recently, the SONIC study found a significant association between higher SBP and lower cognitive function among 70 year olds, while, among 90 year olds, the opposite was found [30]. This study [30] also revealed that, in subjects with hypertension taking antihypertensive medication, a SBP ≥ 140 mmHg might be protective against declining cognitive function in the 90-years age group [30].

### 4.3. Weight Status, Stroke, Cardiac Disease and Cognitive Impairment

Individuals with a history of stroke were at a higher risk of CI than the others in the present study. Post-stroke CI and dementia have been recognized as a major source of morbidity and mortality after stroke [2]. Cognitive impairment and dementia manifesting after a clinical stroke have been categorized as vascular, even in people with comorbid neurodegenerative pathology [12]. Nevertheless, the precise mechanisms underlying a post-stroke worsening of cognitive function are not well established.

Although obesity is a risk factor for stroke in several studies [31], the individuals with CI, in the present study, were more likely to have a history of stroke but also less likely to be obese.

But differential effects of BMI on domain-specific cognitive outcomes after stroke have also been reported [32], particularly that being underweight negatively affected global cognition after an ischemic stroke and also that there was an association between a higher BMI and a significantly worse frontal lobe dysfunction, specifically phonemic and semantic word fluency [32].

Cognitive decline has been found in many heart conditions [33], including heart failure [1], or following coronary artery bypass grafting surgery [3]. We noted a non-significant higher prevalence trend of major cardiovascular disease combining CAD, heart failure and history of stroke in individuals with CI compared to those without CI, *p* = 0.091. Additionally, cardiovascular conditions can expose individuals to the occurrence of strokes, which can lead to cognitive impairment.

A loss of weight is common in stroke occurences and may be related to dysphagia or other neurologic deficits that contribute to eating difficulties. Unintentional weight loss or a low BMI has been defined as an indicator of malnutrition in stroke patients [34] and might be involved in the relation between stroke and CI. Thus, managing weight status with individualized nutritional supplementation could potentially delay cognitive deficits, particularly among patients who had severe stroke.

### 4.4. Undernutrition and Cognitive Impairment

Our results showed that underweight individuals had a higher odds ratio of prevalent CI than the obese individuals and found certain biological findings in favor of nutritional disorders. In fact, individuals with CI were more likely to have a significant higher prevalence of vitamin D and folates deficiencies and also of low prealbumin levels (<0.20 g/L).

The effects of vitamin D supplementation on dementia incidence have been previously assessed [35] in a large prospective cohort of individuals from the National Alzheimer’s Coordinating Center dataset. In this cohort [35], vitamin D exposure was associated with 40% lower dementia incidence versus no exposure [35]. Using an MRI, it has been reported that higher vitamin D levels were linked to greater brain volumes (e.g., white matter, structures belonging to medial temporal lobe) [36].

The evidence of the association between micronutrient malnutrition (i.e., vitamin D, vitamin B 12, folates, antioxidants, protein and lipids) and CI among older adults was highlighted in a recent review [37]. Oxidative stress injury is recognized as a pathogenic cause of CI [38]. Folate and vitamin B12 deficiencies can increase plasma homocysteine, which contributes to oxidative stress, cerebrovascular lesions and cognitive decline. Plasma homocysteine also plays an important role in the pathogenesis of neurodegenerative diseases [39]. Serum prealbumin concentrations, closely related to early changes in nutritional status [40], is considered as a useful marker to assess protein energy malnutrition in hospitalized patients [40].

### 4.5. Underweight, Sarcopenia and Cognitive Impairment

In the present study, being underweight was associated with prevalent CI and being obese seems to have a protective effect on cognitive function.

It is known that muscle wasting, malnutrition and CI tend to occur concomitantly with aging. A loss of lean mass is accelerated in AD and has been associated with brain atrophy and cognitive performance [41]. Sarcopenia, defined as low muscle mass and strength, has been associated with parietal gray matter volume atrophy [42].

All these findings significantly impact the health of the elderly. In fact, in older individuals, age, CI, decreased skeletal muscle mass and sarcopenia were found to be associated with falls [43], which are a common cause of morbidity and mortality.

The results of a recent longitudinal study, identifying BMI as a dose-dependent related factor for cognitive impairment in older adults [44], should encourage the early identification of patients with underweight and/or nutritional risk or malnutrition in order to take care of them and to reduce the risk of dementia.

### 4.6. Limitations

This study has some limitations, including its cross-sectional design, its relatively small sample size and the fact that the AMTS and measurements of biomarkers were limited to one time-point. In addition, we did not take into account any differences in the education level and socio-economic status of the patients or the medication intake and mental health history, which should be considered to support these results. Possible selection bias may be related to the results of the AMTS score since cognitive impairment may not be identified with clinical testing in highly educated individuals.

Genetic factors might also play a role in the relationships between BMI and cognitive decline. In this line, in a recent study based on the clinical and neuropathological records of the National Alzheimer’s Coordinating Center (NACC) in the United States, the results supported that Apolipoprotein E (*APOE*) genotype, a strong genetic factor for AD, modifies the obesity paradox in dementia [45]. In fact, obesity was found to be associated with cognitive decline in early-elderly, cognitively normal individuals without *APOE*4, especially those with *APOE*2.

## 5. Conclusions

The impact of underweight, overweight and obesity on the risk of dementia remains a topic of debate.

In this study of older adults, aged 55 years and older, in the island of Guadeloupe with a high prevalence of comorbid conditions, we found that being underweight was associated with an increased odds ratio of CI, whereas overweight and obesity were not associated with CI. However, detailed longitudinal studies should be conducted in this population to explore the relationships between changes in BMI and changes in the risk of cognitive impairment.

Prevention strategies, including the monitoring of cognitive function and nutritional status and using screening tests, should be put in place in order to maintain an optimal weight and the ability of older adults to have functional capacity and independence and to promote healthy aging.

## Figures and Tables

**Figure 1 healthcare-12-01712-f001:**
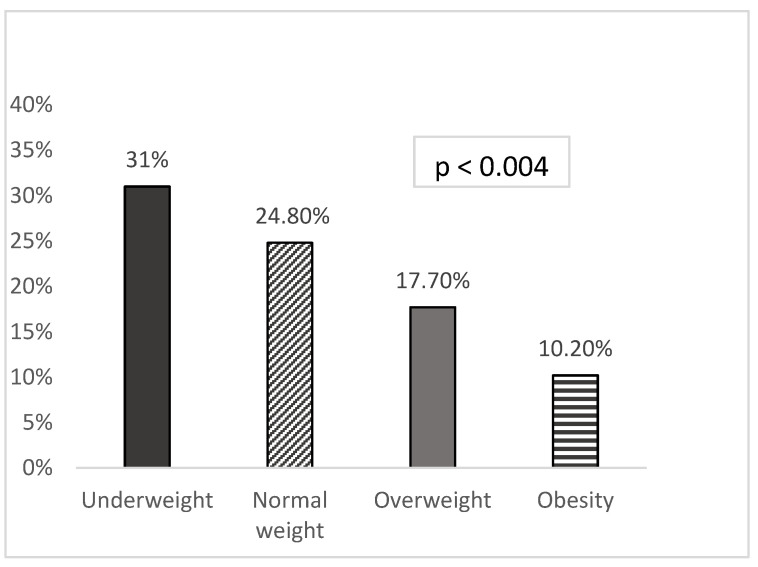
Prevalence of cognitive impairment in underweight (n = 84), normal weight (n = 113), overweight (n = 130) and obese (n = 88) individuals aged 55 years and older.

**Table 1 healthcare-12-01712-t001:** Characteristics of patients according to presence/absence of cognitive impairment.

	Overall		Cognitive Impairment				
				NO		YES	*p*
	N		n		n		
Age (years)	415	75.7 ± 10.1	329	74.7 ± 10.3	86	79.3 ± 8.6	<0.001
Age > 75 years (%)	415	58.8	329	54.1	86	76.7	<0.001
Gender (men) (%)	415	48.2	329	45.3	86	59.3	0.021
Hypertension (%)	415	81.7	329	82.4	86	79.1	0.481
Diabetes-high FBG (%)	415	32.8	329	32.5	86	33.7	0.833
BMI (Kg/m^2^)	415	26.2 ± 6.9	329	26.7 ± 6.7	86	24.4 ± 7.8	0.006
Obesity (%)	415	21.2	329	24.0	86	10.5	0.006
Underweight (%)	415	20	329	17.6	86	29.1	0.018
History of stroke (%)	415	9.6	329	7.9	86	16.3	0.019
Major cardiovascular disease * (%)	415	23.4	329	21.6	86	30.2	0.091
Chronic renal failure (%)	415	41.3	329	39.6	86	47.7	0.155
Nutritional parameters							
Prealbumin level < 0.20 g/L (%)	415	36.1	329	32.5	86	50.0	0.003
Albumin level < 35 g/L (%)	409	40.3	324	38.3	85	48.2	0.096
Folates < 4.89 ng/mL (%)	377	14.6	297	12.1	80	23.8	0.009
25 OH vitamin D < 20 ng/mL (%)	359	12.3	282	10.3	77	19.5	0.029
Vitamin B12 < 200 pg/mL (%)	409	5.1	325	4.3	84	8.3	0.136
Comorbid conditions ≥ 3 ** (%)	415	31.1	329	28.3	86	41.9	0.015

Data are presented as mean ± SD for quantitative variables and as row percentage for qualitative variables. BMI: body mass index. FBG: fasting blood glucose. * Major cardiovascular disease: coronary artery disease, cardiac failure and stroke. ** Comorbid conditions: among 5 parameters: Diabetes–hyperglycemia, hypertension, history of stroke, chronic renal failure and prealbumin level < 0.20 g/L.

**Table 2 healthcare-12-01712-t002:** Prevalence of comorbid conditions according to age categories: age < 75 years and age ≥ 75 years.

	AGE < 75 Years				AGE ≥ 75 Years			
N	All 171	No-CI 151	CI 20	*p*	All 244	No-CI 178	CI 66	*p*
Hypertension (%)	80.7	80.8	80.0	0.933	82.4	83.7	78.8	0.370
Diabetes-high FBG (%)	30.4	28.5	45.0	0.131	34.4	36.0	30.3	0.409
History of stroke (%)	7.6	6.0	20.0	0.026	11.1	9.6	15.2	0.215
Chronic renal failure (%)	30.4	29.8	35.0	0.625	49.0	47.2	51.5	0.548
Prealbumin level < 0.20 g/L (%)	28.7	26.0	50.0	0.025	48.4	38.2	50.0	0.097
Comorbid conditions ≥ 3 ** (%)	24.0	21.2	45.0	0.019	36.1	34.2	40.9	0.337

Data are presented as column percentage for qualitative variables. BMI: body mass index. FBG: fasting blood glucose. Diabetes–high FBG: known diabetes or FBG ≥ 5.6 mmol/L (100 mg/dL). Chronic renal failure: serum creatinine levels > 1 mg/dL (88.40 µmol/L or 10 µg/mL). ** Comorbid conditions: among 5 parameters: diabetes–high FBG, hypertension, history of stroke and chronic renal failure. Prealbumin level < 0.20 g/L.

**Table 3 healthcare-12-01712-t003:** Multivariate logistic regressions of the association between prevalent cognitive impairment and weight status in 415 hospitalized adults aged 55 years and older.

		N 415	Model 1 *			Model 2 **		
			OR	95% CI	*p*	OR	95% CI	*p*
Age ≥ 75 years	No	171	1			1		
	Yes	244	2.54	(1.43–4.49)	0.001	2.40	(1.35–4.27)	0.003
Gender Women		215	1			1		
Men		200	1.72	(1.02–2.88)	0.042	1.62	(0.96–2.74)	0.070
Underweight	No	331	1			----	----	----
	Yes	84	1.85	(1.01–3.39)	0.046	----	----	----
Weight status								
Obesity		88	----	----	----	1		
Overweight		130	----	----	----	1.81	(0.74–4.39)	0.192
Normal weight		113	----	----	----	2.48	(1.03–6.01)	0.044
Underweight		84	----	----	----	3.89	(1.54–9.80)	0.004
Hypertension	No	76	1			1		
	Yes	339	0.88	(0.46–1.70)	0.702	0.94	(0.48–1.83)	0.857
Diabetes-high FBG	No	279	1			1		
	Yes	136	1.15	(0.67–1.99)	0.612	1.22	(0.70–2.13)	0.475
History of stroke	No	375	1			1		
	Yes	40	2.29	(1.10–4.76)	0.028	2.30	(1.09–4.85)	0.030
Chronic renal failure	No	245	1			1		
	Yes	170	1.29	(0.75–2.20)	0.358	1.28	(0.75–2.21)	0.367
Prealbumin level < 0.20 g/L	No	265	1			1		
	Yes	150	1.78	(1.07–2.97)	0.028	1.70	(1.02–2.85)	0.044

Both models were adjusted for age, gender, hypertension, diabetes–high FBG, history of stroke, chronic renal failure, prealbumin level < 0.20 g/L. OR (95% CI): odds ratio (95% confidence interval). * Model 1: with 2 categories for weight status (non-underweight/underweight) and with non-underweight as the reference group. ** Model 2: with 4 categories for weight status (obesity/overweight/normal weight/underweight) and with obesity as the reference group.

## Data Availability

The data presented in this study are available upon reasonable request from the corresponding author. The data are not publicly available because the data are part of an ongoing study.

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
