# Peer review of "Associations between Cognitive Impairment, Weight Status and Comorbid Conditions in Hospitalized Adults of 55 Years and Older in Guadeloupe"

_healthcare, 2024, doi:10.3390/healthcare12171712_

Round 1

Reviewer 1 Report

Comments and Suggestions for Authors

Dear Authors,

I read your paper with interest.

Below are some suggestions that I hope can help you improve it.

Starting from the abstract, I think the approach should be reviewed without starting from the objectives but from the background to then address the objectives, methodology, results, and conclusions. I would give less importance to statistical analysis in this section.

The introduction section is sparse, does not adequately address the issues in a systematic way, and does not raise the possible consequences of cognitive impairment on the health of the elderly.

In particular, it is surprising why the issue of falls in elderly people with dementia is not addressed, as well as the methods to evaluate these aspects, such as the Coonley scale, which in the various items also considers these aspects. This issue, recognized by the WHO as one of the most current and problematic health issues, was recently addressed in an article that I recommend you take into consideration. DOI: https://doi.org/10.1177/25160435241246344

Moreover, the introduction section does not clearly explain the study's purpose or the gap it intends to fill (except for the one indicated to investigate the aspects of dementia in Guadeloupe, which, if not adequately explained, seems an unclear objective). Furthermore, it does not clearly state the study's primary and secondary objectives or what scientific advances they could bring.

The inclusion criteria are not included in the materials and methods section; please specify in detail.

It is unclear whether the clinical scale used (reference no. 18) has been validated in the country where it was administered for the study, where the official language is French.

Figure 1 is not self-sufficient, and statistical significance is not evident.

Please do not use a capital P in the text to indicate statistical significance.

Tables 1-2 are not self-sufficient. Please review the notes in Table 3 in depth.

In line 181, there is a typo in the age indicated.

The discussions are not adequately addressed, but the results are re-proposed, not commented on, and supported by other scientific articles. Please argue the results obtained.

The conclusions are very poorly written, describe the take home message better.

Kind regards

Author Response

Reviewer 1

Comment 1- Starting from the abstract, I think the approach should be reviewed without starting from the objectives but from the background to then address the objectives, methodology, results, and conclusions. I would give less importance to statistical analysis in this section.

R-  In order to comply with the number of 200 words required by the journal, we added a background sentence (page 1, line 20-24) and reduced the results of the statistical analysis.

Comment2 2- The introduction section is sparse, does not adequately address the issues in a systematic way, and does not raise the possible consequences of cognitive impairment on the health of the elderly.

In particular, it is surprising why the issue of falls in elderly people with dementia is not addressed, as well as the methods to evaluate these aspects, such as the Coonley scale, which in the various items also considers these aspects. This issue, recognized by the WHO as one of the most current and problematic health issues, was recently addressed in an article that I recommend you take into consideration. DOI: https://doi.org/10.1177/25160435241246344

Moreover, the introduction section does not clearly explain the study's purpose or the gap it intends to fill (except for the one indicated to investigate the aspects of dementia in Guadeloupe, which, if not adequately explained, seems an unclear objective). Furthermore, it does not clearly state the study's primary and secondary objectives or what scientific advances they could bring.

The inclusion criteria are not included in the materials and methods section; please specify in detail.

It is unclear whether the clinical scale used (reference no. 18) has been validated in the country where it was administered for the study, where the official language is French.

- The "Introduction" paragraph has been modified and we have taken into account the reviewer's comments regarding

- the study’s purpose in the abstract and the introduction              page 1 lines 20-23 and 79-81.

- the study's objectives and scientific advances they could bring.  page 1  lines  75-77 and 79-81

- the inclusion / exclusion criteria,   page 2 lines 84-90

The possible consequences of cognitive impairment on health of the elderly are now more detailed.

Concerning the issue of falls and the reference recommended by the reviewer

R-  We agree with the fact that the  CONLEY Scale is a widespread fall-risk screening tools in medical unit. But, we cannot present it since we did not use it in the present study in which patients were hospitalized between 2021 and 2023 

Nonetheless, we highlighted the problematic of falls associated with cognitive impairment in the discussion of the revised version   page 10, line 407 

 Concerning the clinical scale (AMTS),

R- The AMTS is widely used. The French version of AMTS has been validated by the GERONTONET group

This scale was used in the following French INCUR study:

“Incidence and economical effects of pneumonia in the older population living in French nursing homes: design and methods of the INCUR study.  Demougeot L, et al. BMC Public Health. 2013 Sep 17;13:861.”

-Comment 3 on:  Figure 1 is not self-sufficient, and statistical significance is not evident.

              R- We completed the title of the figure and added the p-value for statistical significance

              Page 5

--Comment 4: Please do not use a capital P in the text to indicate statistical significance.

R- We modified the p for the p-value for statistical significance throughout the manuscript

--Comment 5 Tables 1-2 are not self-sufficient. Please review the notes in Table 3 in depth.

R- We have completed the titles and notes for tables table 1-2-3

--Comment 6: In line 181, there is a typo in the age indicated.

            R- We corrected the age, page 7, line 264 

--Comment 7: The discussions are not adequately addressed, but the results are re-proposed, not commented on, and supported by other scientific articles. Please argue the results obtained.

            R-  We modified the discussion and the results, in the revised version, are more discussed and supported by other articles.    

--Comment 8; The conclusions are very poorly written, describe the take home message better.

              R-  The conclusion has been rewritten   

Reviewer 2 Report

Comments and Suggestions for Authors

Dear Authors,
Thank you for the opportunity to review your manuscript” Associations between Cognitive Impairment, Weight Status and Comorbid Conditions in Hospitalized Adults of 55 Years and Older in Guadeloupe.” Guadeloupe, a French overseas region, is an island group in the southern Caribbean Sea.

This is an interesting study of a particular area’s patient groups, and this may form some level of selection bias. There is also no reference to the nutritional status of the participants and potential living standards relating to the specific location of the study. This may be of importance when analyzing the outcomes.

Line 61-62 you state” i) to estimate the prevalence of cognitive impairment (CI) according to weight status and ii) to identify the comorbid conditions associated with cognitive decline.”

Please confirm your aim to be Cognitive Impairment or Cognitive Decline ?? As this does not match the aims in the abstract.

Within the text many points are referred to, however, all references are at the end of the paragraphs. Please ensure that every point is referenced within the text and not at the end of the paragraph as it makes is difficult to cross reference the exact statements.

Within the data collection you did not measure hip to waist ratio. Is there any reason for this?

With the number of participants, you have not completed a G power to determine the predictability of the results and the measure of acceptance?? Please explain.

Within your clinical parameters, did you allow for the impact of particular medications? Please explain as this may impact your overall results.

In table 1 you show the overall results for Men, is there a similar analysis for women??

Please not in line 181 you need to correct the age specific: “In the present study, in hospitalized Caribbean individuals aged 5 and + years, we” please consider as this looks like a mistake.

Within the discussion it is important to focus on the results you have found without introducing, protein levels ect You may wish to make a statement on what your results may indication and the impact of underweight.

Within the conclusion, you may wish to rewrite “In this study of older adults aged 55 years and older, having high prevalence of comorbid conditions, our study found that be underweight was associated with an increased risk of cognitive impairment.” As this needs some minor correction.

Comments on the Quality of English Language

Moderate editing of English language required

Author Response

Reviewer 2

The authors thank the reviewer for the judicious comments

Comment 1 This is an interesting study of a particular area’s patient groups, and this may form some level of selection bias. There is also no reference to the nutritional status of the participants and potential living standards relating to the specific location of the study. This may be of importance when analyzing the outcomes.

R-. As there was missing clinical data to estimate the nutritional status, we chose to take into account biomarkers of nutritional status for this analysis

Comment 2 Line 61-62 you state” i) to estimate the prevalence of cognitive impairment (CI) according to weight status and ii) to identify the comorbid conditions associated with cognitive decline.”
Please confirm your aim to be Cognitive Impairment or Cognitive Decline ?? As this does not match the aims in the abstract

R- We confirm the aim is “to identify the comorbid conditions associated with cognitive impairment.”

     Lines 22 and 80

Comment 3 -Within the text many points are referred to, however, all references are at the end of the paragraphs. Please ensure that every point is referenced within the text and not at the end of the paragraph as it makes is difficult to cross reference the exact statements.

R- We took this remark into account and some points were referenced within the sentences when it was possible.

Comment 4 Within the data collection you did not measure hip to waist ratio. Is there any reason for this?

R- Waist to hip ratio was not measure in the present study but we plan to measure this tool in the next study.

- Comment 5 With the number of participants, you have not completed a G power to determine the predictability of the results and the measure of acceptance?? Please explain.

R- The power of the study was not calculated.

     The small sample size and the cross-sectional design (noted in the limitations of the study) limit the predictability of the results.   .

Comment 6 -Within your clinical parameters, did you allow for the impact of particular medications? Please explain this may impact your overall results.

R- There is increasing evidence that some treatments may adversely impact cognitive functioning, but this was not taken into account in the present study.  .                                                                         

We noted this point in the limitations of the study,   page 10 line 419

-In table 1 you show the overall results for Men, is there a similar analysis for women??

R- In the revised version, page 4 lines 182-183, we presented the prevalence of CI in men and women.

“Prevalence of CI was 20.7 % in the whole study group and 25.5% in men vs 16.3% in women; p = 0.021”.

 page 4,  lines 182-183

These results were discussed and supported by new references in the paragraph “discussion” of the revised version.

Comment 7 -Please not in line 181 you need to correct the age specific: “In the present study, in hospitalized Caribbean individuals aged 5 and + years, we” please consider as this looks like a mistake.

R- We corrected the age, page 7, line 264.

Comment 8 Within the discussion it is important to focus on the results you have found without introducing, protein levels ect You may wish to make a statement on what your results may indication and the impact of underweight.

R- The discussion has been rewritten and this comment has been taken into account.  

Comment 9 Within the conclusion, you may wish to rewrite “In this study of older adults aged 55 years and older, having high prevalence of comorbid conditions, our study found that be underweight was associated with an increased risk of cognitive impairment.” As this needs some minor correction.

R- The conclusion has been rewritten   

Comment 10 Comments on the Quality of English Language : Moderate editing of English language required

R- We paid particular attention to this remark.

Round 2

Reviewer 1 Report

Comments and Suggestions for Authors

Dear authors,

The paper has improved, but the reviewer's suggestions should be thoroughly followed.

Kind regards.

Author Response

“Associations between Cognitive Impairment, Weight Status and Comorbid Conditions in Hospitalized Adults of 55 Years and Older in Guadeloupe”.

Response to Reviewer 1 Comments

1. Summary

2. General Evaluation

Yes

Can be improved

Must be improved

Not applicable

Does the introduction provide sufficient background and include all relevant references?

( )

( )

(x)

( )

Is the research design appropriate?

( )

( )

(x)

( )

Are the methods adequately described?

( )

( )

(x)

( )

Are the results clearly presented?

( )

( )

(x)

( )

Are the conclusions supported by the results?

( )

( )

(x)

( )

3. Point-by-point response to Comments and Suggestions for Authors

Comments 1: Starting from the abstract, I think the approach should be reviewed without starting from the objectives but from the background to then address the objectives, methodology, results, and conclusions. I would give less importance to statistical analysis in this section.

In order to comply with the number of 200 words required by the journal, we added a background sentence (page 1, line 20-24) and reduced the results of the statistical analysis.

Response 1: In order to comply with the number of 200 words required by the journal, we added a background sentence (page 1, line 20-33) and reduced the results of the statistical analysis.

Comments 2:

The introduction section is sparse, does not adequately address the issues in a systematic way, and does not raise the possible consequences of cognitive impairment on the health of the elderly.

In particular, it is surprising why the issue of falls in elderly people with dementia is not addressed, as well as the methods to evaluate these aspects, such as the Coonley scale, which in the various items also considers these aspects. This issue, recognized by the WHO as one of the most current and problematic health issues, was recently addressed in an article that I recommend you take into consideration. DOI: https://doi.org/10.1177/25160435241246344

Moreover, the introduction section does not clearly explain the study's purpose or the gap it intends to fill (except for the one indicated to investigate the aspects of dementia in Guadeloupe, which, if not adequately explained, seems an unclear objective). Furthermore, it does not clearly state the study's primary and secondary objectives or what scientific advances they could bring.

The inclusion criteria are not included in the materials and methods section; please specify in detail.

It is unclear whether the clinical scale used (reference no. 18) has been validated in the country where it was administered for the study, where the official language is French.

Response 2: We have modified the text

2.1 - The "Introduction" paragraph has been modified and we have taken into account the reviewer's comments regarding

- the study’s purpose in the abstract and the introduction     page 1 lines 20-23 and page 2 71 -74.

- the study's objectives and scientific advances they could bring.  page 2  lines  60-74

- the inclusion / exclusion criteria,  page 2 lines 76-83

2.2 - Concerning the issue of falls and the reference recommended by the reviewer

We agree with the fact that the  CONLEY Scale is a widespread fall-risk screening tools in medical unit. But, we cannot present it since we did not use it in the present study in which patients were hospitalized between 2021 and 2023 

Nonetheless, we highlighted the problematic of falls associated with cognitive impairment in the discussion of the revised version   page 10, line 368. 

 2.3 - Concerning the clinical scale (AMTS),

The AMTS is widely used.

The French version of AMTS has been validated by the GERONTONET group

This scale was used in the following French INCUR study:

“Incidence and economical effects of pneumonia in the older population living in French nursing homes: design and methods of the INCUR study.  Demougeot L, et al. BMC Public Health. 2013 Sep 17;13:861.”

Comment 3 :  Figure 1 is not self-sufficient, and statistical significance is not evident.

 Response .3:    We completed the title of the figure and added the p-value for statistical significance, Page 5

Comment 4: Please do not use a capital P in the text to indicate statistical significance.

Response 4 We modified the p for the p-value for statistical significance throughout the manuscript

Comment 5 Tables 1-2 are not self-sufficient. Please review the notes in Table 3 in depth.

Response  5: We have completed the titles and notes for tables table 1-2-3

Comment 6 in line 181n there ps a typo in the age indicated.

Response 6 We corrected the age, page 7, line 232.

Comment 7: The discussions are not adequately addressed, but the results are re-proposed, not commented on, and supported by other scientific articles. Please argue the results obtained.

Response 7: We modified the discussion and the results, in the revised version, are more discussed and supported by other articles.    

Comment 8; The conclusions are very poorly written, describe the take home message better.

Response 8: The conclusion has been rewritten

Response 9 to Comments on the Quality of English Language

Not applicable

10 . Additional clarifications

In the paragraph “References” of this revised version, we have updated the numbers of the references 15, 16, 17 , 18        which become    16, 17; 18 ;19.
